# Challenges in the Diagnosis and Individualized Treatment of Cervical Cancer

**DOI:** 10.3390/medicina59050925

**Published:** 2023-05-11

**Authors:** Melanie Schubert, Dirk Olaf Bauerschlag, Mustafa Zelal Muallem, Nicolai Maass, Ibrahim Alkatout

**Affiliations:** 1Department of Obstetrics and Gynecology, University Hospital of Schleswig Holstein, Campus Kiel, 24105 Kiel, Germany; 2Department of Gynecology with Center for Oncological Surgery, Charité–Universitätsmedizin Berlin, Corporate Member of Freie Universität Berlin, Humboldt-Universität zu Berlin, and Berlin Institute of Health, Virchow Campus Clinic, 13353 Berlin, Germany

**Keywords:** cervical cancer, HPV, nerve-sparing radical hysterectomy, radiochemotherapy, checkpoint inhibitors

## Abstract

Cervical cancer is still the fourth most common cancer in women throughout the world; an estimated 604,000 new cases were observed in 2020. Better knowledge of its pathogenesis, gained in recent years, has introduced new preventive and diagnostic approaches. Knowledge of its pathogenesis has made it possible to provide individualized surgical and drug treatment. In industrialized countries, cervical cancer has become a less frequent tumor entity due to the accessibility of the human papilloma virus vaccination, systematic preventive programs/early detection programs, health care infrastructure and the availability of effective therapy options. Nevertheless, globally, neither mortality nor morbidity has been significantly reduced over the past 10 years, and therapy approaches differ widely. The aim of this review is to address recent advances in the prevention, diagnostic investigation and treatment of cervical cancer globally, focusing on advances in Germany, with a view toward providing an updated overview for clinicians. The following aspects are addressed in detail: (a) the prevalence and causes of cervical cancer, (b) diagnostic tools using imaging techniques, cytology and pathology, (c) pathomechanisms and clinical symptoms of cervical cancer and (d) different treatment approaches (pharmacological, surgical and others) and their impact on outcomes.

## 1. Introduction

Ninety percent of cervical cancers occur in low- and middle-income countries (LMIC). It is the fourth most common cancer in women worldwide after breast, colorectal and lung cancer. In 2020, the World Health Organization (WHO) estimated a prevalence of 604,000 new cases and 342,000 deaths worldwide [1]. The median age of women at the initial diagnosis of cervical cancer is currently 55 years and has decreased by 15 years over the past 25 years. In about 25 percent of cases, cervical cancer occurs in women younger than 35 years of age. It is the most common gynecologic malignancy during pregnancy, with an incidence of 0.1–12:10,000 [2].

The introduction of screening examinations by the Papanicolaou smear (Pap smear) since 1971 and the human papillomavirus (HPV) vaccination since 2006 led to a dramatic reduction in the incidence of cervical intraepithelial neoplasms (CIN) and cervical cancer in industrialized countries. Cervical cancer is now the thirteenth most common cancer in women in developed countries such as Germany [3]. However, the past 10 years have witnessed no significant reduction in mortality or morbidity; therapy approaches are still very diverse [2].

## 2. Pathogenesis and Risk Factors

Eighty percent of cervical cancers are squamous cell carcinomas. However, the incidence of the less common adenocarcinoma of the cervix has been rising over the last decades. Other rare types include adenosquamous, serous papillary and neuroendocrine cervical carcinoma [4].

The carcinogenesis of cervical cancer is considered to be multifactorial. In addition to common cancer risk factors, such as smoking, promiscuity, long-term use of oral contraceptives, low socioeconomic status and immunosuppression caused by infection, such as by the human immunodeficiency virus (HIV) or drug immunosuppressants after organ transplantation, the most relevant factor in the emergence of cervical cancer is HPV [2].

Eighty percent of sexually active women and men are infected by HPV in their lifetimes, but the infection persists in a mere 5–10% and leads to cervical cancer in just 3% [3,5]. A persistent high-risk HPV infection may cause invasive cervical, vulvar, vulvovaginal, penile, anal, oropharyngeal, or head and neck cancer [2]. The developmental phase from HPV infection to cervical cancer is about 20 years [6]. HPV types most commonly responsible for the development of cervical carcinoma include the high-risk HPV types 16, 18, 45, 31, 33, 58, 52, 35, 59, 56, 6, 51, 68, 39, 82, 73, 66 and 70. Cervical cancer arises from the CIN I–III lesions, which are also classified as low-grade squamous intraepithelial lesions (LSIL), high-grade squamous intraepithelial lesions (HSIL) and adenocarcinoma in situ (ACIS) [2]. With an annual regression rate of 15–23% and a regression rate of 55% in 4–6 years, CIN II is less likely to progress to cervical cancer. In contrast, CIN III has an annual progression rate to invasive carcinoma of 0.2–4% [6].

## 3. Diagnostic Investigation and Prevention

The Pap-only test, the Pap-HPV co-test and the high-risk HPV-only test are the three tests commonly performed for the early detection of cervical cancer. A higher sensitivity, reproducibility and safer prolongation of screening intervals have been proven in several studies for the HPV test compared to conventional cytology or colposcopy [7]. The investigations have demonstrated the safe and sensitive effect of self-sampling for the HPV test, as well as its benefits in LMIC due to its easy and convenient use and its physical and emotional comfort [7,8]. Therefore, HPV self-sampling has been included in the WHO guidelines on self-care interventions for health and well-being published in 2021. The detection of CIN and CIN III by PCR-based self-sampling tests has been demonstrated, but these tests have not been established yet [7].

HPV vaccination is an efficient primary prophylaxis of cervical cancer. In 2007, the recommendations of the Standing Commission on Vaccination (STIKO) imported the precautionary HPV vaccination, which currently advocate vaccination for girls between the ages of 9 and 14 years; since 2018, it has also been recommended for boys of these ages [2]. A prophylactic effect of the HPV vaccination with regard to vaccine-type-specific anogenital diseases has also been shown in women and men aged 14–45 years [9]. The Centers for Disease Control and Prevention (CDC) in the United States recommends a routine HPV vaccination at the age of 11 or 12 years, with the possible start of vaccination at the age of 9 years, a catch-up vaccination to the age of 26 years and a possible vaccination for adults from the age of 27 to 45 years [6].

In terms of tertiary prevention, HPV vaccination was reported to be significantly effective after the surgical treatment of patients with CIN I-III lesions; the risk of developing recurrent CIN was reduced by 58.7% [10]. Despite these data regarding efficacy, the provision of the vaccination on a worldwide basis is still very diverse and worthy of improvement. Less than 30% of LMICs have introduced the HPV vaccination, and only about 20% of women in LMICs have ever been screened for cervical cancer. In contrast, high-income countries have more than an 85% uptake on HPV vaccination, and 60% of women in high-income countries have been vaccinated [11].

Due to the high accessibility and acceptance of vaccines in the population as well as school-based vaccination programs, Portugal, Norway, Iceland, Spain, England and Sweden have high vaccination rates (95%, 85%, 88%, 80%, 80% and 80%, respectively, in 15-year-old women). The proportion of vaccinated persons in Germany was 44.6% in 2018, and in the United States it was 58.6% in 2020 [2]. The current standard Gardasil 9^®^ vaccine is a nonavalent vaccine for the oncogenic/high-risk HPV types 16, 18, 31, 33, 45, 52 and 58 and for the non-oncogenic/low-risk HPV types 6 and 11 [2,12]. HPV vaccination can prevent up to 70% of HPV-related cervical cancers and up to 90% of genital warts [2].

A new classification was published by the Fédération Internationale de Gynécologie et d’Obstétrique (FIGO) in 2018, which supplements the bimanual palpation examination with magnetic resonance imaging (MRI) of the pelvis/computed tomography (CT) of the chest, abdomen and pelvis, as well as biopsies for clinical staging. MRI provides the best assessment of local tumor spreading in terms of parametrial infiltration (sensitivity 84% [13]), whereas CT is used to rule out distant metastases. Surgical staging permits the best assessment of lymph node involvement through systematic lymphadenectomy (LNE). Histological staging has been performed via the TNM classification from 2010 onward. Because the current treatment recommendations are based on data derived from the application of the old FIGO classification of 2009, the following recommendations in this review are also based on the old FIGO classification. Table 1 and Figure 1 reflect the most recent FIGO classification of 2018 [14].

## 4. Surgical Treatment

The therapeutic decision for cervical cancer must be made on an interdisciplinary basis involving gynecologic oncologists, radiation therapists, radiologists, and pathologists. A crucial factor is whether the preservation of fertility is desired and possible. The patient’s wishes as well as her general condition, risk factors, menopausal status and life situation must be included in the decision-making process. Surgery and primary radio(chemo)therapy (RCT) are available as curative treatment options. Based on the current recommendations, a multimodal therapy concept should be avoided because of the resulting increase in morbidity. Furthermore, over- or under-therapy should be avoided [2].

The patient’s lymph node status is one of the most important prognostic parameters in cervical cancer. The determination of tumor stage, prognosis and the resulting therapy decision is based on the intraoperative assessment of the lymph node status. Preoperative imaging with CT, MRI or PET-CT has been shown to be inferior to the surgical detection of lymph node metastases [15]. A comprehensive algorithm for tumor staging is shown in Figure 2.

One approach to therapy de-escalation in early cervical cancer is the use of a sentinel node biopsy (SNB). The sentinel technique for cervical cancer is recommended in the primary stage pTIa1 L1 and/or pTIA 2 and stage pTIB1 (≤2 cm), and it consists of combined detection with technetium-99 and blue dye or, more commonly used today, with intraoperative visualization with indocyanine green (ICG, Figure 3). Ultrastaging to detect low-volume nodal metastases (isolated tumor cells (ITCs) and micrometastases) is recommended in these cases. SNB is feasible and provides excellent detection rates and sensitivity. SNB reduces morbidity compared to pelvic LNE, especially the incidence of lower limb lymphoedema [16]. The three large phase III studies, the Phenix, SENTICOL III and SENTIX trials, are currently evaluating this technique prospectively and will provide further evidence in connection with this procedure [17,18,19,20].

In addition to SNB, the sentinel procedure is used as a part of radical systematic LNE to improve the detection of lymph nodes and to avoid the inclusion of lymphatic drainage pathways, which is associated with a higher rate of morbidity. Systematic radical LNE is used to remove all lymph nodes along the vascular pathways of the associated lymphatic drainage area. Fifteen to twenty pelvic lymph nodes and eight to ten para-aortic lymph nodes are considered representative [2].

Data from the Laparoscopic Approach to Cervical Cancer (LACC) trial, a randomized phase III study, especially those obtained in 2018, have called for significant rethinking in the treatment of cervical cancer. The LACC trial showed that patients after laparotomy have a significantly higher rate of disease-free survival (3-year DFS, 97.1% vs. 91.2%; HR) 3.74; 95% CI, 1.63 to 8.58) and a significantly better overall survival (3-year OS, 99.0% vs. 93.8%; HR 6.00, 95% CI, 1.77 to 20.30) compared to patients who undergo minimally invasive surgery [21]. The clear superiority of the abdominal approach in terms of OS and DFS was also evident in a recently published final analysis of the LACC trial [22]. The cause is still largely unexplained, which is why cervical cancer should currently be operated on via laparotomy and, only in exceptional cases and after appropriate explanation, via a minimally invasive procedure [14].

Studies aimed at reproving the safety of minimally invasive surgery for cervical cancer have been initiated and are currently in progress. Among the leading points of criticism of the LACC trial are the use of transcervical uterine manipulators, the lack of proper tumor containment at the time of colpectomy leading to peritoneal contamination, non-comparable operators/expertise, a low prevalence of robotic-assisted radical hysterectomy and the lack of proper preoperative imaging and assessment [23,24,25,26].

Tumor extirpation by means of previous conization [27,28] or closure using a vaginal cuff showed a comparable DFS and OS in patients treated with a laparoscopy and a laparotomy [24,26]. Ronsini et al. showed, in their meta-analysis, that laparo-assisted vaginal hysterectomies (LARVH) for tumors with a maximum diameter of 2 cm do not appear to affect DFS and OS compared to abdominal radical hysterectomies by using a vaginal cuff to prevent the tumor’s spillage and by not using a uterus manipulator. A statement in the subanalysis about tumors > 2 cm could not be made [24].

Comparable OS, DFS and recurrence rates for open and robotic radical hysterectomies were shown in several reports, thus demonstrating the safety of robotic-assisted radical hysterectomies [25,29,30,31,32,33]. Leitao et al. showed, in their recent systematic review and meta-analysis of cancer outcomes, similar OS and DFS rates for robotic-assisted and laparoscopic surgery [1.01 (0.56, 1.80), *p* = 0.98] or open [1.18 (0.99, 1.41), *p* = 0.06] [33]. However, large prospective randomized studies will be needed to cause a change in current guideline recommendations.

The Robot-assisted Approach to Cervical Cancer (RACC) trial, an international, randomized controlled multicenter trial, is currently in progress. Women with early-stage cervical cancer are randomly assigned to robotic-assisted surgery or a laparotomy. The results of this study are eagerly awaited [34]. Likewise, the ongoing Robotic versus Open Radical Hysterectomy for Cervical Cancer (ROCC) trial will provide more detailed insights into this currently critical issue [23].

The different radicality of hysterectomies is classified into the five grades of Piver I–V according to Piver et al., among others, and is listed in Figure 4 [34,35].

### 4.1. Nerve-Sparing Radical Hysterectomy

The goal of surgical treatment in cervical cancer is not only tumor-free resection but also the preservation of organ functionality and crucial nerves. Modern imaging with MRI provides three-dimensional anatomical information about principal nerves such as the hypogastric nerve (HN), the inferior hypogastric plexus (IHP) and the pelvic splanchnic nerve (PSN). Bladder functionality is achieved through sympathetic nerves for bladder relaxation and parasympathetic innervation for bladder contraction. Therefore, the preservation of the HN, the PSNs, the IHP and the vesical nerve branches must be ensured [36]. A detailed anatomical illustration of these essential nerves is shown in Figure 5.

A classical radical hysterectomy involves the complete resection of the cardinal ligament along with pelvic splanchnic nerves. In a nerve-sparing radical hysterectomy, the superior HN in conjunction with sympathetic innervation is visualized at the level of the aortic bifurcation, followed by bilateral dissection along the sigmoid colon. The pelvic splanchnic nerve in conjunction with parasympathetic innervation is exposed and spared from the lateral aspect at the same level. Parametrial dissection during a nerve-sparing radical hysterectomy is performed under directed visualization of the contiguous pelvic autonomic nerves. During resection of the dorsal parametrium, the HN in the mesoureter is spared with a previous preparation. The resection of the ventral and lateral parametrium must be performed under the viewing and sparing of the IHP, the bladder branches of the IHP and the PSN [36,37,38,39].

It should be mentioned here that nerve-sparing radical hysterectomies are not confined to early-stage cervical cancer. Tumor size plays no role when making the decision about sparing the autonomic nerve system, which must be the standard of care even for tumors > 4 cm in size [38,40].

The classic triad of complications following a radical hysterectomy consists of the lack of bladder sensitivity, a hypo- or non-contractile detrusor and disturbed coordination of detrusor contraction; these should be strictly avoided. Rectal nerve preservation can also prevent slow transit constipation. An underestimated complication of radical hysterectomies is the loss of function of the genital cavernous bodies and the associated vaginal lubrication, which can also be reduced by using the nerve-sparing surgical technique. Nerve-sparing surgery was shown to significantly reduce postoperative morbidity and improve quality of life [38,41,42,43].

**Figure 5 medicina-59-00925-f005:**
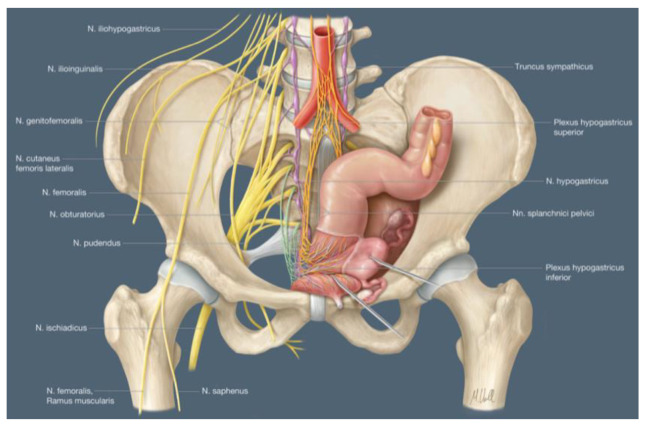
Anatomical illustration of somatic and autonomic pelvic nerves in a female pelvis. Yellow: somatic nerves; orange: sympathetic plexus; purple: sympathetic trunk; green: pelvic splanchnic nerves (PSN), courtesy of Alkatout et al. [44].

### 4.2. Fertility Preservation

As 40% of patients with cervical cancer are of reproductive age, the preservation of fertility must be taken into account if the patient wishes to preserve her fertility and when such preservation is oncologically justifiable. According to the National Comprehensive Cancer Network (NCCN) guidelines, the European Society of Gynecological Oncology (ESGO)/European Society for Radiotherapy and Oncology (ESTRO)/European Society of Pathology (ESP) guidelines, as well as the German guidelines, fertility-sparing treatment can be offered to patients with stage IA1 to IB1 squamous cell carcinoma and adenocarcinoma of the cervix. Fertility-sparing surgery is not recommended for gastric-type adenocarcinoma, small cell neuroendocrine histology and non-HPV-related adenocarcinomas [2,14,45]. The treatment options for preserving fertility include conization; simple and radical trachelectomy performed abdominally, vaginally or by conventional laparoscopy; or robotic assistance with prophylactic permanent cerclage. Whereas conization or simple trachelectomy is performed with pregnancy rates of 71–75% in stage IA1 and IA2 without risk factors, radical trachelectomy with permanent cerclage should be used as a fertility-preserving procedure in stage IA1 L1 V0, stage IA2 V0 or stage IB1 and IIA1 V0 < 2 cm. From stage IA1 L1 onward, prior histopathological exclusion of lymph node metastases with SNB or LNE is mandatory [2,14,46]. Neoadjuvant chemotherapy can be administered in stage IB1 ≥ 2 cm, followed by conization or trachelectomy within studies; this procedure is not recommended as a standard therapy [2].

The selection of patients suitable for fertility preservation should be made on the basis of the pathological extent/stage of the tumor, as well as the patient’s comorbidities and her likelihood of becoming pregnant and carrying the pregnancy to term. The patient must be informed in detail about the risks, advantages, and therapy alternatives in a strict sense of shared decision making. Therefore, multidisciplinary consultation with gynecologic oncologists, fertility specialists, pathologists, radiologists and radiation therapists is essential before making therapy decisions. The oncologic safety of these techniques has been proven, but patients must be educated about the fact of a high-risk pregnancy with a higher risk of miscarriage, preterm birth (31–57%) and primary cesarean section after trachelectomy. Similarly, cervical stenosis (5–15%) with associated infertility may occur postoperatively. A hysterectomy need not be performed routinely after completed family planning, but it is recommended in the case of HPV persistence, Pap abnormality, a desire for maximum safety, or limited or abolished accessibility of the cervix [2,46,47].

### 4.3. Stage-Specific Therapy Guides

The following therapy algorithms for the individual stages of cervical cancer are listed according to the German guidelines [2], the NCCN [46] and the ESGO/ESTRO/ESP guidelines [14].

Patients with stage IA1 without a risk factor should be treated with conization and cervical curettage or a simple hysterectomy in cases of positive margins after conization, completed family planning or a desire for greater safety. In the case of positive margins after conization and the desire to preserve fertility, the patient may be offered re-conization or trachelectomy with prophylactic permanent cerclage. A secondary hysterectomy can be performed in this setting, as mentioned before [2,14,46].

In stage IA1, with the invasion of lymphatic vessels (L1), SNB should be performed in accordance with the same therapy recommendations as those of stage IA1 without L1. SNB is also indicated in stage IA1 with at least two risk factors and in stage IA2 with one risk factor. The involvement of SLN is an indication for systematic LNE followed by RCT, as well as prior ovariopexy with bilateral salpingectomy to preserve intrinsic ovarian function in premenopausal patients. Piver I is performed in cases of disease-free pelvic SLN.

Fertility cannot be preserved in stage IA2 with at least two risk factors. SNB is performed as a part of surgical staging in this setting. In the case of negative SLN, the approach used here is Piver II with bilateral salpingo-oophorectomy if needed [2].

In contrast, the NCCN guidelines for stage IA1 with L1 and stage IA2 recommend a modified radical hysterectomy after previous pelvic LNE or SLN mapping, or in cases of inoperable patients/stage IA2 pelvic external beam radiotherapy (ERBT) with brachytherapy [46]. The ESGO/ESTRO/ESP guidelines recommend adjuvant radiotherapy alone in stage 1A2 with L1, stage 1B1 or stage 2A1 [14].

The international guidelines recommend the use of the Sedlis criteria as a guide for adjuvant treatment decisions in node-negative, margin-negative and parametria-negative cases. The Sedlis criteria include greater than one-third stromal invasion, capillary lymphatic space involvement and a cervical tumor diameter greater than 4 cm [46,48].

In the case of diseased pelvic lymph nodes, surgical staging is again extended to include para-aortic LNE to remove the affected lymph nodes and to determine the radiation field for subsequent RCT [2,14].

Patients with stage IB1 < 2 cm without risk factors should be treated with a radical hysterectomy, Piver II, in the case of negative pelvic SLN [14]. If the patient wishes to preserve her fertility, it would be advisable to perform a radical trachelectomy with prophylactic permanent cerclage. Surgery should only be performed in the absence of an upfront indication for adjuvant radiotherapy [2]. The NCCN or ESGO/ESTRO/ESP guidelines provide no recommendations for a routine secondary hysterectomy [14,46].

Patients with stage IIA1 should be treated with a radical hysterectomy, Piver II, with a tumor-free resection margin of the vaginal cuff. In postmenopausal patients or premenopausal patients with adenocarcinoma, the German guidelines recommend a radical hysterectomy, Piver III, with a tumor-free margin of the vaginal cuff and bilateral salpingo-oophorectomy for stage IB2, IIA2 and IIB with a maximum of two risk factors. RCT is recommended in stage 1B2 and higher with positive margins or residual tumors, including positive lymph nodes on imaging [2,14]. In contrast, the NCCN guidelines give preference to RCT over surgery in stage IB2 and IIA2. The German and ESGO/ESTRO/ESP guidelines advocate RCT rather than surgery in stage IIB and above [2,14]. Whereas the German guidelines recommend surgical lymph node staging via pelvic and para-aortic LNE for stage IB2 and IIA2 tumors with the possible removal of affected lymph nodes, the NCCN guidelines recommend only lymph node staging via radiologic imaging in these stages [2,46].

The recommendation for stage III is surgical staging via systematic LNE with the removal of malignant lymph nodes or radiological assessment of malignancy prior to R(C)T [2]. The ESGO/ESTRO/ESP guidelines advocate for the possibility of para-aortic LNE, at least up to the inferior mesenteric artery with negative para-aortic lymph nodes on imaging and debulking of suspicious pelvic lymph nodes, whereas the NCCN guidelines recommend radiologic staging for stage IB2 and higher [14,46].

In stage IVA, the choices, among others, are R(C)T or primary exenteration in selected cases. In stage IVB, the focus is on symptom-oriented therapy, which consists of radiotherapy or RCT, palliative chemotherapy combined with bevacizumab and with or without pembrolizumab, or best supportive care [2,14,49,50]. Therapy algorithms for the individual tumor stages are shown in Figure 6, adapted according to the German guidelines.

## 5. Radio(Chemo)Therapy

Intensity-modulated radiotherapy, individualized MRI-guided brachytherapy or image-guided adaptive brachytherapy (IGABIT) should be used in the primary RCT of cervical cancer; this approach provides optimal protection of surrounding tissue by reducing gastrointestinal and urogenital toxicities as well as acute and late therapy-related reactions. The approach also permits the safe use of selective dose escalation or a simultaneous integrated boost [2]. RCT is given with cisplatin 40 mg/m^2^ body surface area for 5 weeks as a radiosensitizer and may be administered as a neoadjuvant, primary or adjuvant therapy. In the event of existing contraindications to chemotherapy with cisplatin, such as renal failure, radiation alone is used. Previous LNE may help to define the radiation field, which extends over the pelvic and para-aortic lymphatic drainage area and is referred to as extended-field radiotherapy in the sense of percutaneous radiotherapy when para-aortic lymph nodes are also affected. Brachytherapy must be included if the hysterectomy is not a part of the primary operative therapy/staging procedure. Percutaneous radiation of the primary tumor and lymph nodes is primarily performed with combined cisplatin-containing chemotherapy, followed by brachytherapy of the primary tumor [2,14,46]. Adjuvant RCT should be given to patients with histologically confirmed postoperative high-risk factors, such as lymph node metastases and parametrial infiltration, as well as positive resection margins. Adjuvant radiotherapy alone is the adjuvant therapy of choice for intermediate-risk factors or positive Sedlis criteria, such as lymphovascular space invasion, invasion of more than a third of cervical stromal and tumor size > 4 cm [51]. Clearly, the high morbidity of adjuvant RCT after surgery should be considered and discussed with the patient [46].

## 6. Medical Therapy

In addition to surgical and radiologic (lymph node) staging, the current standard therapy for locally advanced cervical cancer is primary simultaneous RCT with cisplatin [2,14,46]. Patients with recurrent, persistent or metastatic cervical cancer should be treated with a combination of cisplatin/topotecan or cisplatin/paclitaxel and bevacizumab. This combination has shown benefits in terms of survival (8.2 months vs. 6 months; HR 0.68 [95% CI 0.56–0.84]; *p* = 0.0002), DFS (13.3 months vs. 16.8 months; HR 0.77 [95% CI 0.062–0.95]; *p* = 0.007) and response rates (49% vs. 36%; *p* = 0.003) in a GOG 240 study [45,50]. A randomized phase III JCOG0505 trial showed equal efficacy; carboplatin can be used instead of cisplatin to prevent nephrotoxicity and neutropenia [52]. More recent data from the phase II CECILIA trial showed that the combination of carboplatin with paclitaxel and bevacizumab achieves comparable efficacy and has a favorable side effect profile [53]. Nab-paclitaxel, vinorelbine, ifosfamide, topotecan, pemetrexed or irinotecan can be used in second-line therapy [2].

Analogous to the treatment of endometrial cancer, new data have been obtained for cervical cancer in regard to immune checkpoint inhibitors. The largely HPV-dependent carcinogenesis of cervical cancer explains the high immunogenicity of these tumors. Thus, immunotherapy could be successful in this setting. The programmed cell death ligand PD-1/PD-L1 system is particularly important for the course of the disease and is involved in carcinogenesis. A KEYNOTE 158 trial revealed a response rate of 15% and promising OS rates (median, 9.4 months for the entire study population and 11.0 months for the PD-L1-positive population) in patients with advanced cervical cancer and PD-L1 positivity of tumor tissue. This is roughly comparable to established therapies. What is outstanding, however, is the favorable response rate within 2.1 months and the pronounced duration of the response [54].

A randomized, placebo-controlled, double-blind phase 3 (KEYNOTE 826) trial yielded notable results. Adding pembrolizumab (200 mg every 3 weeks), a humanized monoclonal PD-1 antibody, to cisplatin or carboplatin/paclitaxel with or without additional bevacizumab, a recombinant monoclonal antibody to human vascular endothelial-derived growth factor (VEGF), in patients with advanced cervical cancer and a positive PD-L1 status (combined positive score (CPS) ≥ 1) demonstrated a significant improvement in median OS (not reached vs. 16.3 months; HR 0.64; *p* = 0.0001) and median DFS (10.4 vs. 8.2 months; HR 0.62; *p* < 0.0001). Compared to the placebo group, the treatment with pembrolizumab yielded an objective response rate of 68% vs. 50%, as well as a median response duration of 18.0 vs. 10.4 months [49,55]. Based on these data, the Food and Drug Administration (FDA) approved pembrolizumab 2021 as first-line therapy in combination with chemotherapy, with or without bevacizumab, for the treatment of patients with persistent, recurrent or metastatic cervical cancer whose tumors express PD-L1 (CPS ≥ 1). Pembrolizumab has also been approved as a sole therapy in patients with recurrent or metastatic cervical cancer who are progressive on chemotherapy.

Currently, the effect of pembrolizumab in addition to concurrent RCT in primary advanced cervical cancer is being investigated in an ENGOT-cx11/KEYNOTE-A18 trial. Completion of the trial is anticipated in December 2024 [56].

Nivolumab (anti-PD-1) and ipilimumab (cytotoxic T-lymphocyte antigen 4 antibody; anti-CTLA4) are other forward-looking compounds. Monotherapy with nivolumab and in combination with ipilimumab are being tested in the ongoing CheckMate 358, a multicenter, multicohort phase I/II trial. The treatment response ranged between 26.3% (nivolumab treatment, 95% CI, 9.1 to 51.2) and 38.4% (nivolumab and ipilimumab) in non-pretreated patients. Even patients with prior lines of treatment achieved a response rate of 34.9%. These studies showed that, in patients who responded, the response was of a rather long duration compared to that of conventional therapy. The median OS was 21.6 months (95% CI 8.3–46.9), 15.2 months (95% CI 9.0–36.2) and 20.9 months (95% CI 14.4–32.8), and the median PFS was 5.1 months (95% CI 1.9–9.1), 3.8 months (95% CI 2.1–10.3) and 5.8 months (95% CI 3.8–9.3) in patients treated with nivolumab monotherapy, a combination of nivolumab 3 mg/kg and ipilimumab 1 mg/kg every 2 weeks, and a combination of nivolumab 1 mg/kg and ipilimumab 3 mg/kg every 6 weeks, respectively [57].

Another promising antibody tested and proven in an EMPOWER-Cervical 1/GOG-3016/ENGOT-cx9 study, an open-label, multicenter, phase 3 trial, is cemiplimab (anti-PD-1), which most recently achieved a 3.5 month improvement in OS with an acceptable side effect profile as a monotherapy, regardless of PD-L1 status, in advanced cervical cancer [58]. Cemiplimab was recently approved as a monotherapy by the European Medicines Agency in patients with recurrent or metastatic cervical cancer whose disease progressed on or after platinum-based chemotherapy.

Tisotumab vedotin, another new substance, is a tissue factor-directed antibody–drug conjugate (ADC) and showed good efficacy in a multicenter, phase II innovaTV 204/GOG-3023/ENGOT-cx6 trial; further testing in phase III studies will be needed [59]. The therapy mechanism of tisotumab vedotin is based on its complex formation with tissue factors and the subsequent intracellular release of monomethyl auristatin E, a microtubule-disrupting agent, which leads to cell cycle arrest and apoptotic cell death.

In addition to the mentioned immunotherapies for cervical cancer, further promising immunotherapy strategies such as the engineered T cell receptor (TCR)-like antibody therapy and engineered chimeric antigen receptor (CAR) T cell therapy have yielded promising outcomes in terms of cytotoxicity and cervical tumor regression. Resistance to T-cell-mediated recognition, toxicity, patient specificity (genetically engineered T cells, no standard treatment) and high costs are current hurdles that still need to be overcome [60,61].

### Neoadjuvant Chemotherapy

Currently, neoadjuvant chemotherapy (NACT) is not a standard of care in cervical cancer but may result in operable findings in selected patient. Although no improvements have been shown yet in PFS and OS, the following advantages have been noted: potential preservation of fertility, a significantly lower incidence of lymph node metastases and parametrial invasion and a lesser need for adjuvant RCT [62,63,64]. Therefore, platinum-based NACT followed by a radical hysterectomy represents a therapy option, especially for selected patients with stage Ib2-IIb. However, the side effects of chemotherapy must be weighed against its benefits. Trifanescu et al., in a retrospective study of 108 patients treated with neoadjuvant radiotherapy ± chemotherapy followed by robotic-assisted radical hysterectomy, reported a pathological complete response rate of 66% and a DFS of 100% at 36 months in early-stage patients with stage IB-IIA, and 80% in advanced-stage stage IIB-IVA [32]. Nevertheless, compared to cisplatin-based RCT, this treatment option is inferior in OS and PFS. The number of patients who remain inoperable and require definitive chemoradiation is about 25–30%. The value of NACT compared to cisplatin-based RCT remains a subject of ongoing investigation [65,66,67,68].

## 7. Conclusions

Important data have been obtained recently about the prevention, diagnosis and treatment of cervical cancer. The three pillars of therapy are surgery, RCT and medical therapy. The effectiveness of the HPV vaccination has been established. However, we face barriers in the achievement of high vaccination rates, especially in LMICs.

Screening programs for cervical dysplasia and cervical cancer have become widely established. Rhythmic screening for HPV and the PAP smear, if necessary, with a colposcopy for further clarification, have already been successful in terms of minimizing cervical dysplasia. However, the effective implementation of these programs in LMICs—countries with the highest incidence of cervical cancer—remains unresolved. The results of the LACC trial, which yielded proof of a higher rate of recurrence, a lower rate of DFS and a lower OS among patients undergoing minimally invasive surgery, have led clinicians to reconsider their view about the surgical approach.

Immune checkpoint inhibitors in particular could serve as a relatively well-tolerated therapy option, with a very prolonged response in patients with advanced cervical cancer whose disease prognosis is predictably poor.

The goal is not only to generate global accessibility for the effective prevention of cervical cancer, but also to include patients in randomized, prospective trials and to standardize evidence-based recommendations for all patients in order to improve disease-free survival, overall survival and quality of life.

## Figures and Tables

**Figure 1 medicina-59-00925-f001:**
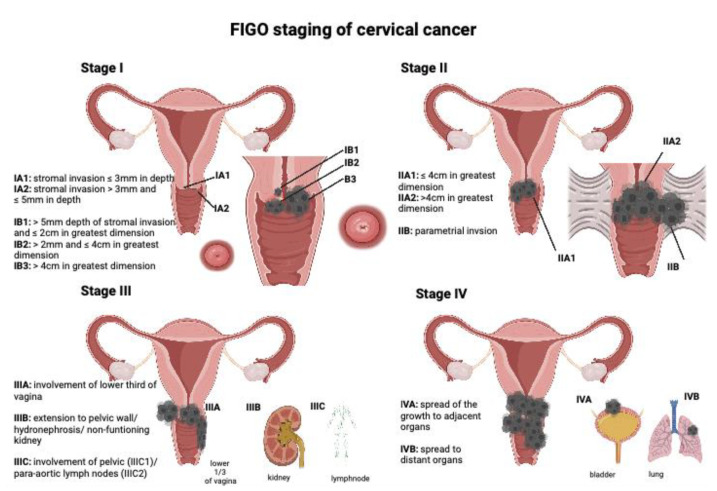
FIGO staging system created with BioRender.com.

**Figure 2 medicina-59-00925-f002:**
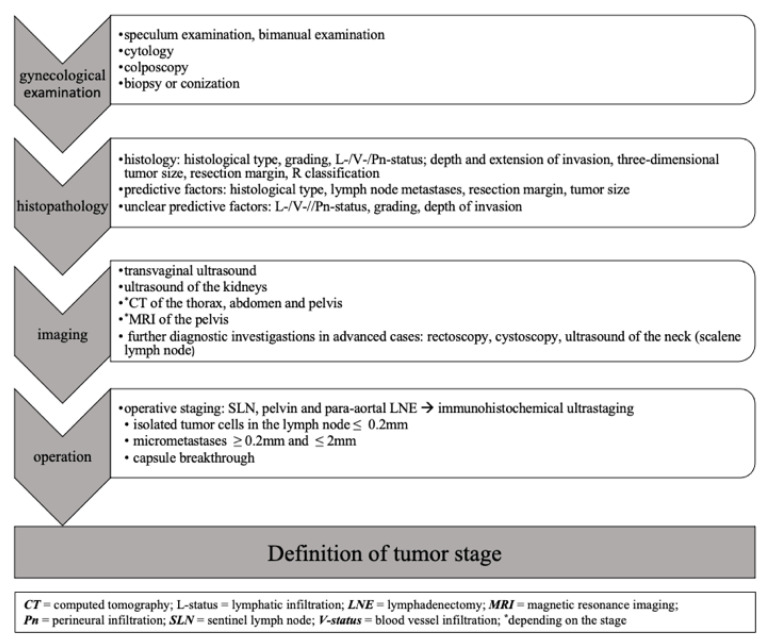
Algorithm of tumor stage definition in cervical cancer, adapted from the German S3 guidelines [2].

**Figure 3 medicina-59-00925-f003:**
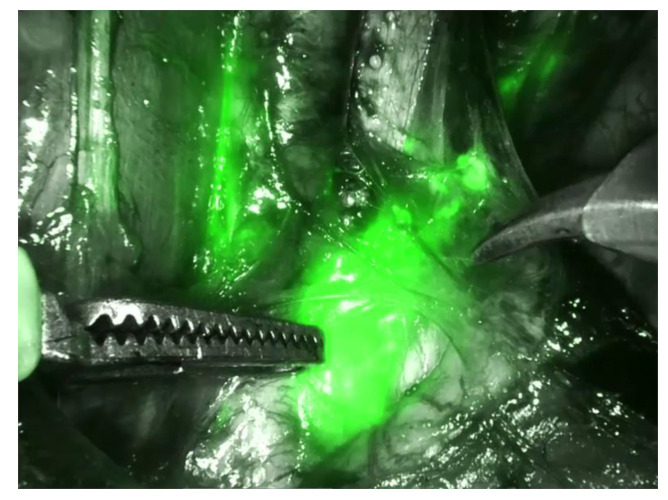
Intraoperative sentinel node visualization with indocyanine green at the Department of Obstetrics and Gynecology, University Hospital of Schleswig Holstein, Campus Kiel, Germany.

**Figure 4 medicina-59-00925-f004:**
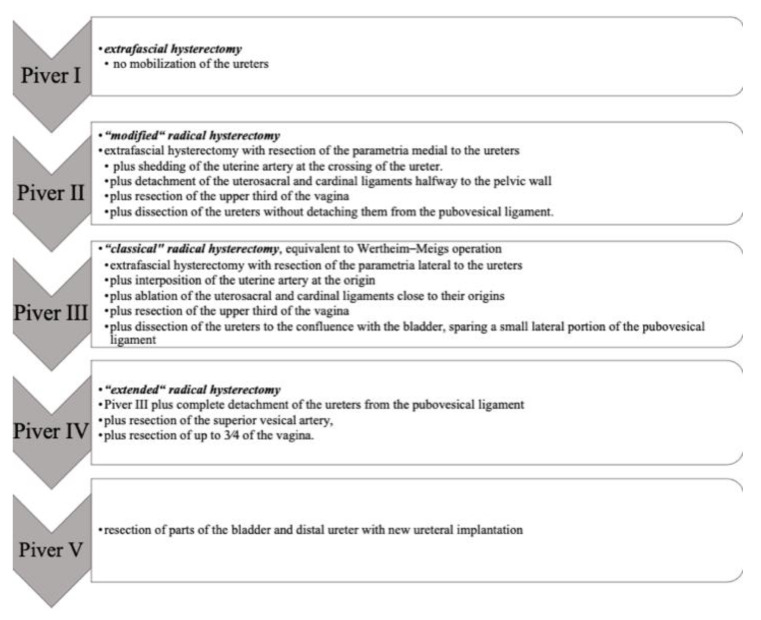
Classification of hysterectomy classified by Piver et al. [35].

**Figure 6 medicina-59-00925-f006:**
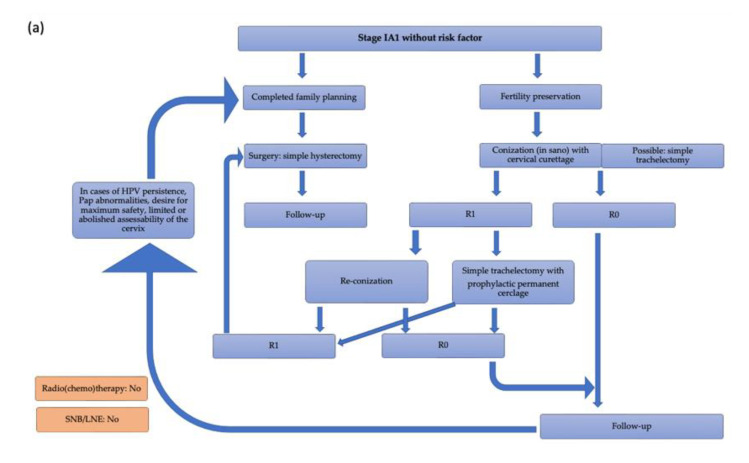
Therapy algorithm for cervical cancer based on the German guidelines: (**a**) Stage IA1 without risk factors; (**b**) Stage IA1 with L1; (**c**) Stage IA1 with ≥2 risk factors and stage IA2 with 1 risk factor; (**d**) Stage IA2 with ≥2 risk factors; (**e**) Stage IB1, IIA1; (**f**) Stage IB2, IIA2, IIB with ≤2 risk factors; (**g**) Stage III, stage IVA, stage IVB [2].

**Table 1 medicina-59-00925-t001:** FIGO classification 2018 [14].

Stage			
I			Carcinoma strictly confined to the cervix
	IA		Invasive carcinoma with maximum depth of invasion ≤ 5 mm
		IA1	Stromal invasion ≤ 3 mm in depth
		IA2	Stromal invasion > 3 mm and ≤5 mm in depth
	IB		Deepest invasion > 5 mm; lesion limited to cervix uteri with size measured according to maximum tumor diameter
		IB1	>5 mm depth of stromal invasion and ≤2 cm in greatest dimension
		IB2	>2 cm and ≤4 cm in greatest dimension
		IB3	>4 cm in greatest dimension
II			Invasion beyond the uterus, but no extension into the lower third of the vagina or to the pelvic wall
	IIA		Involvement limited to the upper two thirds of the vagina without parametrial invasion
		IIA1	≤4 cm in greatest dimension
		IIA2	>4 cm in greatest dimension
	IIB		Parametrial invasion but not to the pelvic wall
III			Involvement of the lower third of the vagina and/or extension to the pelvic wall and/or causes hydronephrosis or non-functioning kidney and/or involvement of pelvic and/or para-aortic lymph nodes
	IIIA		Involvement of the lower third of the vagina, with no extension to the pelvic wall
	IIIB		Extension to the pelvic wall and/or hydronephrosis or non-functioning kidney (unless known to be due to another cause)
	IIIC		Involvement of pelvic and/or para-aortic lymph nodes (including micrometastases), irrespective of tumor size and extent
		IIIC1	Pelvic lymph node metastases only
		IIIC2	Para-aortic lymph node metastases
IV			Extension beyond the true pelvis or involvement (biopsy proven) of the mucosa of the bladder or rectum
	IVA		Spread of the growth to adjacent organs
	IVB		Spread to distant organs

## Data Availability

The datasets analyzed for the current study are available from the corresponding author upon reasonable request.

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
