# Peer review of "Challenges in the Diagnosis and Individualized Treatment of Cervical Cancer"

_medicina, 2023, doi:10.3390/medicina59050925_

Round 1
Reviewer 1 Report
The authors reviewed recent progress in the diagnosis, prevention, and therapy of cervical cancer. More details about different treatment options, especially different surgical options were addressed. The review will help clinicians to be updated on the latest progress in the treatment of cervical cancer.
Minor points:
1. With the success of CAR-T therapy in blood tumors, the immunotherapy of tumors has entered a new era. Immunotherapy of cervical cancer, such as the therapeutical vaccines and TCR-T cell targeting HPV proteins, should be included in the review as well.
2. In line 420, “cemiplimab” is actually an antibody targeting PD-1 which should be mentioned in the text. And so, in line 422, “regardless of PD-1 status” should be “regardless of PD-L1 status”.
3. In line 426, an antibody-drug conjugate “tisotumab vedotin” is described. To help the readers understand how the drug works in cervical cancer treatment. More information about the target of tisotumab should be provided.
Author Response
Specific comment of reviewer 1:
Comments and Suggestions for Authors:
The authors reviewed recent progress in the diagnosis, prevention, and therapy of cervical cancer. More details about different treatment options, especially different surgical options were addressed. The review will help clinicians to be updated on the latest progress in the treatment of cervical cancer.
Minor points:
- With the success of CAR-T therapy in blood tumors, the immunotherapy of tumors has entered a new era. Immunotherapy of cervical cancer, such as the therapeutical vaccines and TCR-T cell targeting HPV proteins, should be included in the review as well.
- In line 420, “cemiplimab” is actually an antibody targeting PD-1 which should be mentioned in the text. And so, in line 422, “regardless of PD-1 status” should be “regardless of PD-L1 status”.
- In line 426, an antibody-drug conjugate “tisotumab vedotin” is described. To help the readers understand how the drug works in cervical cancer treatment. More information about the target of tisotumab should be provided.
Response to Reviewer 1 Comments:
Thank you very much for your interest in this work and for taking the time to make detailed comments. In the revised version we have made the following changes:
- We added therapeutic vaccines and TCR-T cell targeting HPV proteins as a new therapy aspect (pages 14-15, line 435-444)
- We have specified the exact substance group of cemilimab as well as mentioned the correct indication (page 14, line 423, 425).
- We added a detailed description of the mechanism of action of tisotumab (page 435, line 429-434)

Reviewer 2 Report
Thank you for submitting you valuable work. Cervical cancer is a major health problem especially in less evolved countries.
In my opinion the review needs some salt and pepper. I recommend to extend the part regarding following topics
1. Surgical treatment. I will focus more on minimally invasive surgical technics and I recommend the authors explanation way Ramirez trial (published also in NEJM) was not positive.
2. Robotic surgery is fashionable nowadays and I recommend more focus on this topic, even if the results of randomized trial are not available considering that there ar a lot of articles on Pubmed regarding this topic (UwinsMedlin, Trifanescu, The RECOURSE Study: Long-term Oncologic Outcomes Associated With Robotically Assisted Minimally Invasive Procedures for Endometrial, Cervical, Colorectal, Lung, or Prostate Cancer: A Systematic Review and Meta-analysis Mario M Leitao.
3. A new chapter regarding neoadjuvant treatment may be included.
4. References are a little old and not enough for a comprehensive review.
5. Language must be corrected (recommending are recommending 277 and many others)
Author Response
Specific comment of reviewer 2:
Comments and Suggestions for Authors:
Thank you for submitting your valuable work. Cervical cancer is a major health problem especially in less evolved countries.
In my opinion the review needs some salt and pepper. I recommend to extend the part regarding following topics
- Surgical treatment. I will focus more on minimally invasive surgical technics and I recommend the authors explanation way Ramirez trial (published also in NEJM) was not positive.
- Robotic surgery is fashionable nowadays and I recommend more focus on this topic, even if the results of randomized trial are not available considering that there ar a lot of articles on Pubmed regarding this topic (Uwins, Medlin, Trifanescu, The RECOURSE Study: Long-term Oncologic Outcomes Associated With Robotically Assisted Minimally Invasive Procedures for Endometrial, Cervical, Colorectal, Lung, or Prostate Cancer: A Systematic Review and Meta-analysis Mario M Leitao.
- A new chapter regarding neoadjuvant treatment may be included.
- References are a little old and not enough for a comprehensive review.
- Language must be corrected (recommending are recommending 277 and many others)
Response to Reviewer 2 Comments:
Thank you very much for your interest in this work and for the time and effort taken to make constructive suggestions. Please note the following changes:
- Based on your recommendation, we have described in detail the data published by Ramirez on the LACC trial with respect to the criticisms and the publications cited in this regard (page 6, line 170-181).
- Additionally, we have highlighted robotic surgery as another point of focus to be discussed in cervical cancer (page 6, line 170-187).
- We have added a new chapter on neoadjuvant therapy in cervical cancer. Nevertheless, we continue to focus on the current guidelines.
- We have included more recent references.
- The review was sent for English editing and revised.

Round 2
Reviewer 2 Report
I thanks the authors for there modification, I belive that the article is suitable for publication